# A Robust Discriminant Framework Based on Functional Biomarkers of EEG and Its Potential for Diagnosis of Alzheimer’s Disease

**DOI:** 10.3390/healthcare8040476

**Published:** 2020-11-11

**Authors:** Qi Ge, Zhuo-Chen Lin, Yong-Xiang Gao, Jin-Xin Zhang

**Affiliations:** Department of Medical Statistics, Sun Yat-sen University, Guangzhou 510080, China; geqi3@mail2.sysu.edu.cn (Q.G.); linzhch3@mail2.sysu.edu.cn (Z.-C.L.); gaoyx8@mail2.sysu.edu.cn (Y.-X.G.)

**Keywords:** Alzheimer’s disease, diagnosis, electroencephalography, MODWT, functional biomarker

## Abstract

(1) Background: Growing evidence suggests that electroencephalography (EEG), recording the brain’s electrical activity, can be a promising diagnostic tool for Alzheimer’s disease (AD). The diagnostic biomarkers based on quantitative EEG (qEEG) have been extensively explored, but few of them helped clinicians in their everyday practice, and reliable qEEG markers are still lacking. The study aims to find robust EEG biomarkers and propose a systematic discrimination framework based on signal processing and computer-aided techniques to distinguish AD patients from normal elderly controls (NC). (2) Methods: In the proposed study, EEG signals were preprocessed firstly and Maximal overlap discrete wavelet transform (MODWT) was applied to the preprocessed signals. Variance, Pearson correlation coefficient, interquartile range, Hoeffding’s D measure, and Permutation entropy were extracted as the input of the candidate classifiers. The AD vs. NC discriminant performance of each model was evaluated and an automatic diagnostic framework was eventually developed. (3) Results: A classification procedure based on the extracted EEG features and linear discriminant analysis based classifier achieved the accuracy of 93.18 ± 3.65 (%), the AUC of 97.92 ± 1.66 (%), the F-measure of 94.06 ± 4.04 (%), separately. (4) Conclusions: The developed discrimination framework can identify AD from NC with high performance in a systematic routine.

## 1. Introduction

Alzheimer’s disease (AD) is an irreversible neurodegenerative disorder and the leading cause of dementia, which slowly destroys patients’ memory, thinking, and eventually, the ability to conduct basic bodily functions [1]. The progressive accumulation of disability affects the daily lives of patients, their families, and wider society [2]. The number of dementia patients is drastically increasing with the increasing life expectancy and the rapidly aging global population; 132 million people are expected to be affected in 2050 [3]. None of the treatments available today can prevent or block AD’s progression. The most effective method for preserving brain function is believed to be based on early diagnosis and active management started from the very beginning of the clinical spectrum of the disease [1]. The pathophysiological process of AD begins 20 years or more before noticeable symptoms such as memory loss and cognitive dysfunction [4,5]. Further, the AD symptoms are in many cases confused with a normal aging process, thus usually resulting in a late diagnosis when the disease is practically irreversible [6,7]. Early diagnosis is crucial for optimal treatment planning and may avoid rapid deterioration of the illness, thus improving the life quantity of both patients and caregivers. These have driven the search for early diagnosis biomarkers towards the timely detection of this insidious disease [8,9,10].

Evidence suggests that amyloid β protein (Aβ) accumulation and neuronal degeneration (elevated tau/phosphorylated tau) are two major biomarkers of AD, and also the core neuropathological elements of the disease [11]. The former is thought to develop first during the AD continuum, indicating the initiating events from decades prior to clinical symptoms, which can be measured in vivo by positron emission tomography (PET) amyloid imaging and low cerebrospinal fluid (CSF) Aβ42 [10]. The latter is shortly before the appearance of the clinical symptoms, with elevated CSF tau, decreased fluorodeoxyglucose (FDG) uptake on PET, and atrophy on structural magnetic resonance imaging (MRI) [12]. While CSF and neuroimaging markers are gold standards for the in vivo assessment of the patients, they are invasive and expensive and, therefore, have limited utility as frontline screening and diagnostic tools for AD [13]. Moreover, the recent study shifts focus on biomarkers that reveal AD-associated disturbances in brain function rather than simply in structure. Further, there is growing evidence that synaptic loss is most closely correlated with cognitive impairment, suggesting measures of brain functional deficits as early biomarker candidates of AD [14,15]. Electroencephalography (EEG) is a widely available and noninvasive diagnostic method that reflects the real-time functioning of brain synapses [16]. Quantitative EEG (qEEG) analysis provides multi-perspective information of EEG signals such as its frequency components (spectral domain), dynamic alterations (time domain), and source imaging (spatial domain). Numerous studies have demonstrated that qEEG as potential markers for brain function evaluation can detect the following abnormalities in AD patients: (a) modifications in EEG rhythms, (b) decrease in synchronization, (c) reduced complexity, and (d) neuromodulatory deficits [17,18]. Moreover, qEEG provides objective and quantifiable results that can be repeated in further studies, and also has the advantages of requiring simpler testing procedures and lower cost [14,19]. These make it optimal for large-scale screening and early diagnosis of AD. To this end, EEG as an appropriate tool for analysis of AD has been extensively studied in the last decades [17]. Nevertheless, as far as we know, few of the research results help clinicians in their everyday practice and decision support. The problem lies in the fact that the EEG signal is susceptible to noise and has characteristics of being non-stationary, making diagnosis challenging. Moreover, as a large variability exists - in inter-subject- [20,21], it is hard to discriminate artifacts and patterns form normal brain activity. Reliable biomarkers and systematic diagnostic procedures with the capability to extract useful information from rough EEG signals are still urgently required [22,23].

Wavelet Transform (WT) has been suggested as an effective time–frequency analysis method for EEG signal processing [24,25]. It is to convolve the EEG signal with a time window with adjustable width, using small window widths at higher frequencies and large window widths at low frequencies. This adapts well to the characteristics of EEG signals that are composed of short duration events of high-frequency and low-frequency events of long duration [26]. Time-frequency analysis of EEG signals integrated with Machine Learning (ML) techniques may promote a deeper comprehension of AD and aid the diagnosis [27,28,29]. Machine learning algorithms, such as feature selection (FS), removing redundant information from the high dimensional data, could avoid over-fitting and improve diagnosis precision. Motivated by the encouraging outcomes obtained with the WT analysis and ML methods [23,29,30], the study aims to explore robust functional biomarkers based on time-frequency features of qEEG and develop a computer-aided discriminant system for classifying EEG signals of AD and normal elderly controls (NC) automatically. In this study, EEG signals were preprocessed by down-sampling, filtering, segmenting, manual inspection, and independent components analysis firstly to get relatively clean epochs, and maximal overlap discrete wavelet transformation (MODWT) were applied to obtain the time-frequency information of EEG signals. Then, four wavelet features (Variance, Pearson correlation coefficient, interquartile range, Hoeffding’s D measure) and Permutation entropy, which aim at characterizing the change of complexity and synchronization of EEG due to AD, were extracted per channel per epoch. The 20 most discriminant features, selected by dimensionality reduction technique, were fed into conventional ML-based classifiers to realize automatic identification of AD from NC. The results showed that the proposed EEG features and framework could be used for diagnosis and disease progression monitoring of AD with high performance.

## 2. Materials and Methods

The workflow of the proposed discriminant framework is summarized in Figure 1. It mainly involves four steps: (1) EEG data acquisition and preprocessing; (2) Maximal overlap discrete wavelet transform; (3) Feature extraction and selection; (4) Classification and validation. The details were represented in this section.

### 2.1. EEG Data Acquisition

A total of 46 eligible subjects were enrolled between September 2012 and September 2019 from the First Affiliated Hospital of Sun Yat-sen University (SYSU) in Guangzhou (China) and their raw EEG data were collected. Ethics approval was obtained from the Ethics Committee of School of Public Health of SYSU (No.123), as well as consent from the participants in the study. The subjects comprised two groups: the first group (NC) consisted of 23 normal elderly controls without memory loss complaints and functional cognitive decline (14 females, aged 59 (50, 74) years), and the second group (AD) of 23 patients diagnosed with AD symptoms (12 females, aged 73 (56, 80) years). The diagnosis of AD patients were based on the Aging and the Alzheimer’s Association (NIA-AA) criteria [31]. The details were that: (1) the core clinical presentation of AD such as evident cognitive or behavioral (neuropsychiatric) symptoms were met through history-taking and the Mini Mental State Examination (MMSE), and (2) neuroimaging tools such as the structural or functional (fMRI) scans were performed for detecting AD biomarkers and excluding other brain diseases, and (3) excluding patients with other forms of dementia or accompanying Parkinson’s disease, epilepsy, psychiatric disorders, and serious organic disease.

All subjects were righthanded and homogeneous for sex and age; indeed, the Chi-square test did not show statistically any difference between the sex of two groups (*χ*^2^ = 0.35, *p* = 0.552). In addition, the two sided Mann–Whitney *U* test did not find any statistical difference in the age of the two groups (*U* = −183.50, *p* = 0.075). The details of the participants’ characteristics are shown in Table 1. The sample size was augmented to 631 epochs (NC epochs of 237 and AD of 394) by segmenting the continuous signals into smaller epochs introduced later in Section 2.2; this data augmentation process was commonly used in previous works [23,32,33].

Resting-state EEG was recorded using Nicolet EEG machine with 500 or 125 Hz sampling rate. Scalp electrodes were placed according to the international 10–20 montage system, and Cz electrodes were used as reference. Sixteen-channel EEG signals (Fp1, Fp2, F3, F4, C3, C4, P3, P4, O1, O2, F7, F8, T3, T4, T5, and T6) were acquired with the subjects awake, relaxed, and with their eyes closed for at least five minutes. The subjects were readily prompted if the EEG technicians who continuously monitored the EEG traces detected any signs of drowsiness. Under the circumstances, the brain background activity was recorded.

### 2.2. EEG Preprocessing

As EEG is prone to a variety of artifacts, the step of preprocessing and denoising is necessary before analysis. Firstly, EEG recordings with sampling frequency of 500 Hz were down-sampled to 125 Hz for the convenience of subsequent processing. Following, low EEG frequency artifacts, like those generated by eye blinking and movements, were attenuated by applying to all signals a high pass filter at 0.5 Hz using the *pop_eegfiltnew* function of the *EEGLab* toolbox in Matlab, and power grid interference of 50 Hz were removed by applying notch filter using *filtfilt* function. Afterward, every signal was segmented into *s* non-overlapping EEG epochs sized *l* = 10 s (where *s* depends on the EEG length) and baseline drift was removed. An overall of 631 epochs was finally obtained. The segmenting step not only augments the sample size but also assures that each piece-wise signal (epoch) is quasi-stationary [34]. The acquired EEG signals were later visually inspected to identify every 10-s epochs free of excessive artifacts. If there were less than three channels with excessive artifacts, they were then interpolated using spherical interpolation, otherwise the segments were rejected. The person performing the preprocessing was blinded to whether the participants were NC or AD. Lastly, automated artifact removal was performed on the manually processed dataset using the wavelet-enhanced independent components analysis (wICA) algorithm [35]. This automated enhancement method was shown to be the best for AD assessment [36]. The above was performed using the EEGLAB and *wICA* toolboxes in Matlab R2014a software, resulting in the preprocessed EEG dataset.

### 2.3. Maximal Overlap Discrete Wavelet Transform (MODWT)

The maximal overlap discrete wavelet transform (MODWT) is a modified version of the discrete wavelet transform (DWT) [37,38]. Both of them are defined in terms of the pyramid algorithm that decomposes the original signal into different layers by wavelet and scaling filters; each layer represents different frequency components of the input signal. The DWT requires the length of input time series must be an integer power of 2 and it is sensitive to the starting point of the time series due to the down-sampling. The MODWT eliminates this down-sampling and creates wavelet coefficients of which the length at each layer will be the same as the original time series, thus it is more applicable in most contexts.

Let the DWT wavelet and scaling filter be
$\left\{h_{j,l}\right\}$ and
$\left\{g_{j,l}\right\}$, *l* = 0, 1, 2, …, *L* − 1, where *L* is the filter width and *j* the layer of decomposition. The MODWT wavelet $\left\{{\tilde{h}}_{j,l}\right\}\;$ and scaling filters $\left\{{\tilde{g}}_{j,l}\right\}$ are directly defined by renormalizing the DWT filters: ${\tilde{h}}_{j,l}=h_{j,l}/2^{j/2}$ and ${\tilde{g}}_{j,l}=g_{j,l}/2^{j/2}$. Scaling filter ${\tilde{g}}_{l}$ is a quadrature mirror filter that corresponds to wavelet filter ${\tilde{h}}_{l}$ by the equation ${\tilde{g}}_{l}={\left(-1\right)}^{l+1}{\tilde{h}}_{L-1-l}$; the inverse relationship is ${\tilde{h}}_{l}={\left(-1\right)}^{l}{\tilde{g}}_{L-1-l}$. Then, the MODWT wavelet, ${\tilde{W}}_{j}$, and scaling, ${\tilde{V}}_{j}$, coefficients of layer *j* are defined as the convolution of the time series *X* = $\left\{X_{t},\;t=0,\;1,\;2,\ldots ,N-1\right\}$ and the MODWT filters [39]:(1)$${\tilde{W}}_{j,t}=\sum_{l=0}^{L_{j}-1}{\tilde{h}}_{j,l}X_{t-l\;\mathrm{mod}\;N},$$
(2)$${\tilde{V}}_{j,t}=\sum_{l=0}^{L_{j}-1}{\tilde{g}}_{j,l}X_{t-l\;\mathrm{mod}\;N},$$
where $L_{j}=\left(2^{j}-1\right)\left(L-1\right)+1$. From the above expressions, the MODWT wavelet coefficients is well-defined for any sample size *N*.

The wavelet detail (series of inverse wavelet coefficients), ${\tilde{D}}_{j}$, and smooth (the inverse of the series of scaling coefficients), ${\tilde{A}}_{j}$, at *j*th layer are defined as:
(3)$${\tilde{D}}_{j,t}=\sum_{l=0}^{N-1}{\tilde{h}}_{j,l}{\tilde{W}}_{j,t+l\;\mathrm{mod}\;N},$$
(4)$${\tilde{A}}_{j,t}=\sum_{l=0}^{N-1}{\tilde{g}}_{j,l}{\tilde{V}}_{j,t+l\;\mathrm{mod}\;N}.$$

The original time series can be decomposed from the multi-resolution analysis (MRA) of the MODWT as follows:(5)$$X={\tilde{D}}_{1}+{\tilde{D}}_{2}+\cdots +{\tilde{D}}_{j}+{\tilde{A}}_{j}.$$

The choice of wavelet filters is important in conducting wavelet analysis. Different wavelet families have a trade-off between the degree of symmetry (i.e., linear phase characteristics of wavelets) and the degree to which ideal high-pass filters are approximated [40]. The selected wavelet should be well adapted to the events analyzed. Here, wavelet filters of length 2 of Haar, of lengths 4, 6, and 8 of the Daubechies family (D4, D6, and D8), of length 8 from the least asymmetric (LA8), and of length 6 from the Coiflets family (C6) were used as candidates to generate the MODWT coefficients, and hence the MODWT features.

### 2.4. Feature Extraction and Selection

Given an EEG epoch (*n* × L, with *n* = 16 channels, L = 1250 samples), the MODWT was taken on each channel of the epoch, according to Equations (1) and (2), coming up with 16 × **j* wavelet coefficient per epoch, where *j* is the number of layer of decomposition depending on the selected wavelet filter. For *j*th wavelet coefficient, the following features are estimated:Variance (*VA*):
(6)$$V_{X,j}^{2}=\frac{1}{M_{j}}\sum_{t=L_{j}-1}^{N-1}W_{j,t},$$
where $M_{j}\equiv N-L_{j}+1$.

Pearson correlation coefficient (*PCC*):

(7)$$\rho_{XY,j}=\frac{\sum_{t=L_{j}-1}^{N-1}\left(W_{X,j,t}-{\bar{W}}_{X,j}\right)\left(W_{Y,j,t}-{\bar{W}}_{Y,j}\right)}{\sqrt{V_{X,j}}*\times \sqrt{V_{Y,j}}},$$

Interquartile range (*IQR*):

(8)$$IQR_{X,j}=P_{75}\left(W_{X,j,t}\right)-P_{25}\left(W_{X,j,t}\right),$$

Hoeffding’s *D* measure (*D*):

(9)$$D=30\left(\frac{\left(N-2\right)\left(N-3\right)D_{1}+D_{2}-2\left(N-2\right)D_{3}}{N\left(N-1\right)\left(N-2\right)\left(N-3\right)\left(N-3\right)}\right),$$
where, $D_{1}=\sum_{i=1}^{N}Q_{i}\left(Q_{i}-1\right)$, $D_{2}=\sum_{i=1}^{N}R_{i}\left(R_{i}-1\right)\left(R_{i}-2\right)\left(S_{i}-1\right)\left(S_{i}-2\right)$, $D_{3}=\sum_{i=1}^{N}\left(R_{i}-2\right)\left(S_{i}-2\right)Q_{i}$; The $R_{i}$ and $S_{i}$ are ranks of the epoch $W_{X,j,t}$ and $W_{Y,j,t}$, the $Q_{i}$ is the bivariate ranks.

Besides the above four time-frequency features based on wavelet coefficients, permutation entropy as a useful feature based on the rank of original signals was also extracted.

Permutation entropy (*PE*):

(10)$$PE=\frac{-\sum_{j=1}^{m!}p_{j}\ln\left(p_{j}\right)}{\ln\left(m!\right)},$$
where, $p_{j}=\frac{n_{j}}{\sum_{j=1}^{m!}n_{j}}$, *m* is the embedding dimension, $n_{j}$ is the number of times the *j*th permutation is occurring.

Table 2 shows the maximum number of decompose layer for the six filters selected in Section 2.3, the layer of 1–4 carrying nonsense information is excluded in this study. The total numbers of the features obtained for each epoch based on different filters are also showed in Table 2 (16 (# channel) × *j* (# layer) × 2 (for *VA* and *IQR*) + 120 (# combination of any two channels) × *j* (# layer) × 2 (for *PCC* and *D*) + 16 (for *PE*)).

High-dimensionality feature vectors often lead to bias and overfitting due to redundant information, especially when limited data is available. The stepwise implementation of [41] was used to perform feature reduction, i.e., to select the top 20 features for each classification algorithm.

### 2.5. Classification and Validation

A total of eight supervised learning classifiers were employed to identify EEG epochs as belonging to AD or HC subjects. They are linear discriminant analysis (LDA), logistic regression (Logreg), k-nearest neighbor (KNN), support vector machine (SVM), random forest (RF), naive Bayes (Nbayes), adabag boosting (Adaboost), and neural network (NNet). Classifier performance was evaluated using the five-fold cross-validation (CV5), which is particularly useful for small sample sizes. In CV5, The EEG epochs were initially divided into 5 folders, and 4 folders were then selected as training set and the remaining folder as test set. This step was repeated 5 times until each folder has been used as a test set. The dividing results of the sample set by CV5 were saved and used for each model evaluation for comparison. It should be noted here that all epochs of a subject were fixed as a “blocking” that would not be separated when dividing the folders. In this way, we were sure that each subject’s epochs were not simultaneously present in the training and test groups in each iteration, thus reducing over-fitting and avoiding misleading results. Hyperparameters optimization was also performed to limit overfitting and increase model generalizability.

For evaluating the classification performances of each model, the following metrics were adopted:
*Accuracy*, or correct rate:
(11)$$Accuracy=\frac{TP+TN}{TP+FP+TN+FN},$$

*Precision*, or Positive Predictive Value:

(12)$$Precision=\frac{TP}{TP+FP},$$

*Recall*, True Positive Rate (*TPR*) or sensitivity:

(13)$$Recall=\frac{TP}{TP+FN},$$

*Specificity*, True Negative Rate (*TNR*):

(14)$$Specificity=\frac{TN}{FP+TN},$$

*F-measure*:

(15)$$F-measure=\frac{2\times Recall\times Precision}{Recall+Precision},$$

Area under the curve (*AUC*) for the receiver operating characteristic curve:

(16)$$AUC=\frac{\sum_{ins_{i}\in positiveclass}rank_{ins_{i}}-\frac{M\times \left(M+1\right)}{2}}{M\times N},$$
where, *M*, *N* are the number of positive sample and negative sample, separately. *True Positives* (*TP*): the number of correctly classified AD; *False Positives* (*FP*): the number of incorrectly classified NC; *True Negatives* (*TN*): the number of correctly classified NC; *False Negatives* (*FN*): the number of misclassified AD.

Although accuracy is an intuitive measure used in several contexts, just getting a high accuracy is not enough to ensure good classification performance, since it is possible that it hides a low number of *TP* or *TN* [42]. To counter this problem, we also calculated the *F-measure* and *AUC*. A high value of *F-measure* implies that both Precision and Recall are close to 1, meaning the model performed well. The *AUC* can reflect the total effect of Sensitivity and Specificity.

The process, including classification, parameters tuning, and model validation, were achieved using R package “mlr”.

## 3. Results

### 3.1. Selection of Wavelets Filters

As described in Section 2.3, six wavelet filter candidates, including Haar, D4, D6, D8, LA6, and C6, were used for the wavelet-based procedures to generate the MODWT coefficients, and hence the MODWT features. Figure 2 shows the AD vs. NC discriminant performances of the six filters based on LDA in terms of six evaluation metrics. As can be seen, the LA8 filter performed best, i.e., all six evaluation measures of LA8 were the highest. The wavelet features got from LA8 were used in the following analysis. It should be noted here that the LA8 also performed well when using other classifiers (See Appendix A for more details).

### 3.2. Discrinimant Performance of Different Classifiers

Table 3 reports the AD vs. HC discriminant performances using the eight classifiers (linear discriminant analysis (LDA), logistic regression (Logreg), k-nearest neighbor (KNN), support vector machine (SVM), random forest (RF), naive bayes (Nbayes), adabag boosting (Adaboost), and neural network (NNet)) in terms of six model evaluation metrics. The five-fold cross-validation resulted in five subsets from the original dataset, so all classification measures are herein expressed in terms of mean value ± standard deviation (i.e., mean (Accuracy) ± std (Accuracy)) over five outputs.

It is noticeable that linear discriminant analysis is superior to all other classifiers mentioned here, i.e., the value of six evaluation measures from LDA are all higher than the other classifiers. Indeed, LDA reports the accuracy of 93.18 ± 3.65 (%), the AUC of 97.92 ± 1.66 (%), the F-measure of 94.06 ± 4.04 (%), the specificity of 91.45 ± 6.98 (%), the recall of 94.55 ± 3.85 (%), and the precision of 94.02 ± 7.95 (%). Besides, the results of Logreg and NNet is close to LDA, they give the second and the third highest accuracy of 92.44 ± 3.61 (%) and 91.47 ± 4.93 (%), separately. Other classifiers performed less well than these three, the contrast can be seen clearly in Figure 3. Moreover, the standard deviation of accuracy of LDA is similar to that of Logreg, and smaller than NNet, showing a relatively shorter boxplot in Figure 3, indicating the discriminant performance of LDA is stable in the 5 subsets of CV5.

### 3.3. Discrinimant Performance of Different Features

Furthermore, we are motivated to examine the inclusion of the five features to assess whether they provide useful information. The individual classification results for each feature as input to LDA are reported in Table 4, and this can be seen intuitively in Figure 4. The proposed parametric features (the combination of *VA* and *PCC*) and the nonparametric features (the combination of *IQR*, *D*, and *PE*) gave similar performance when discriminating NC and AD patients, showing the accuracy of 89.52 ± 6.32 (%) and 88.83 ± 4.18 (%), respectively. The *PCC* and *D* also gave relatively high accuracy of 88.18 ± 5.28 (%) and 84.75 ± 3.77 (%), compared with *VA* (71.32 ± 7.25 (%)), *IQR* (70.32 ± 9.38 (%)); and *PE* showed the lowest accuracy of 57.09 ± 19.75 (%) between the five features. The combination of all five features achieved the best accuracy of 93.18 ± 3.65 (%). The other five model evaluation measures gave similar comparison results.

## 4. Discussion

EEG recordings, which are non-invasive, cheap, and acceptable by the aging, is an ideal diagnostic tool for AD. Early detection of AD based on EEG biomarkers has been extensively studied. However, reliable markers extraction for better generalization remains challenging due to the noisy, nonlinear, non-stationary, and multidimensional nature of EEG signals [43]. The study aims to develop a robust discriminant framework based on quantitative EEG biomarkers by combining mature signal processing methods with modern machine learning techniques. Results show that the proposed routine is effective for differential diagnosis of AD patients and NC with an average accuracy of 93.18%.

An appropriate wavelet filter can well adapt the feature of the events present in signals, thus maximizing the information extracted [38]. The LA8 wavelet gives a moderately higher accuracy compared with the other five wavelets used in this study (Figure 2). The LA8 with a relatively large length improves the approximation to an ideal high-pass filter and can better match the characteristic features in the signals, thus, it can give suitable MODWT feature for detection of AD.

The linear classifier is the best choice in this research framework compared with other more complex ML classifiers mentioned here (Table 3 and Figure 3). This is not surprising, although EEG signals have the characteristics of randomness and non-stationarity, which are difficult to be processed by normal linear classifiers. In this study, before automatic classification, MODWT was used to decompose the signals in a locally nonlinear way, then the discriminative wavelet features were extracted, making the discriminant much easier [44]. Figure 5 shows that the input features satisfy the assumption of Gaussian distribution and are linearly separable. The classification results show quite favorable accuracy of 93.18 ± 3.65 (%) and AUC of 97.92 ± 1.66 (%) when using the LDA with all features together.

With multidimensional time series, such as 16 channels for an epoch here, the discriminant task is more complex because as well as taking into account the multiple series associated with each subject, it is of great interest to exploit the relationships between the components of each channel [45,46,47]. In this analysis the *PCC* and *D* of wavelet coefficients are used to describe the quantitative relationship between the multi-channels, reflecting the character of brain synchronization. When using each feature as input variable, the accuracy of 88.18 ± 5.28 (%) of *PCC* was the highest and the accuracy of 84.75 ± 3.77 (%) of *D* was the second highest between the five features (Table 4 and Figure 4). It is clearly an indication that these interrelationship features provide useful information about the inter-channel relations and hence make an important contribution to distinguishing between the patterns of EEG signals of individuals with AD, and those of NC. While PE gave the lowest accuracy of 57.09 ± 19.75 (%) when used alone, this may because that one epoch only has 16 PE features from 16 channels, extracted from the original series, not the transformed wavelet coefficient, resulting in the information from *PE* being much less than the other four wavelet features. The *VA* and *IQR* could assess the change of brain complexity in AD patients and can provide supplementary information for discrimination. The best classification results were obtained when all five features were used. Further, the proposed parametric features (the combination of *VA* and *PCC*) and the nonparametric features (the combination of *IQR*, *HCC*, and *PE*) exhibited similar performance.

In conclusion, a robust discriminant framework based on functional biomarkers of qEEG used to identify Alzheimer’s disease patients from normal elderly controls was developed in this study. The time-frequency analysis method of MODWT accompanying with LA8 wavelet better captured the characteristic features in the EEG signals of AD and thus well adapted to the classification task. The combination of five extracted features with the LDA classifier achieved the best average accuracy of 93.18%, the AUC of 97.92%, the F-measure of 94.06%, the specificity of 91.45%, the recall of 94.55%, and the precision of 94.02%, separately. This is an inspiring result because the features selected in the study and classifier of linear discriminant analysis are easy to calculate and interpret, indicating quantitative EEG combining signal processing techniques and machine learning methods could be a powerful tool. Further, the EEG with 16 channels used in this study were widely available in low-income and middle-income countries, making the proposed classification framework more practical. In a word, the proposed discriminant framework can achieve an automatic diagnosis of AD with high accuracy in a systematic way, as well as aid AD screening.

One of the major limitations of the current study is the small sample size used to train and test the model, but the data augmentation and five-fold CV compensate for this. By segmenting the continuous EEG signals, we finally obtained 631 epochs for analysis, and the cross-validation ensured the stability of the classification results. Larger studies are needed to generalize the findings of the study in the future. Moreover, in future work, EEG data of early patients, such as mild cognitive impairment (MCI) patients who have not yet developed into AD, need to be collected to verify whether the proposed framework could realize earlier detection of possible AD patients and perform a tri-classification task (AD vs. MCI vs. NC). Most of the patients came to the First Affiliated Hospital of SYSU (one of the best hospitals in South China) for severe clinical symptoms, which resulted in the absence of the MCI in this study. The gradual realization of medical information sharing in the future could help us track the information of patients with Alzheimer’s disease at early stages or before they become ill, thus promoting the early diagnosis of the disease. However, the classification framework proposed in the study based on the existing study still has potential for aiding the diagnosis of AD with high performance and can be easily reproduced in future data.

## Figures and Tables

**Figure 1 healthcare-08-00476-f001:**
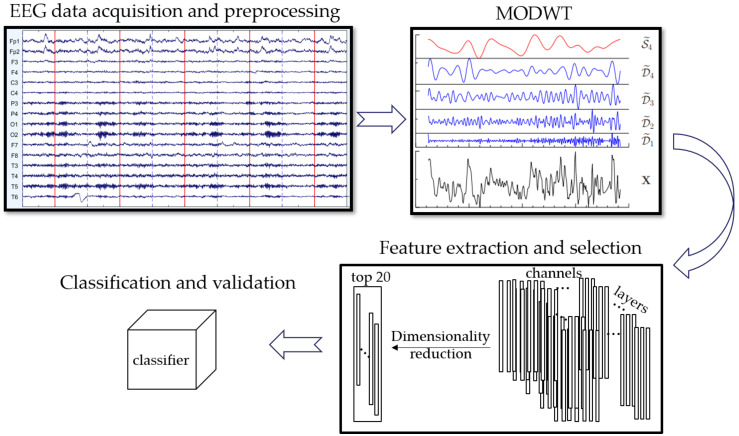
The flow diagram of the discriminant framework.

**Figure 2 healthcare-08-00476-f002:**
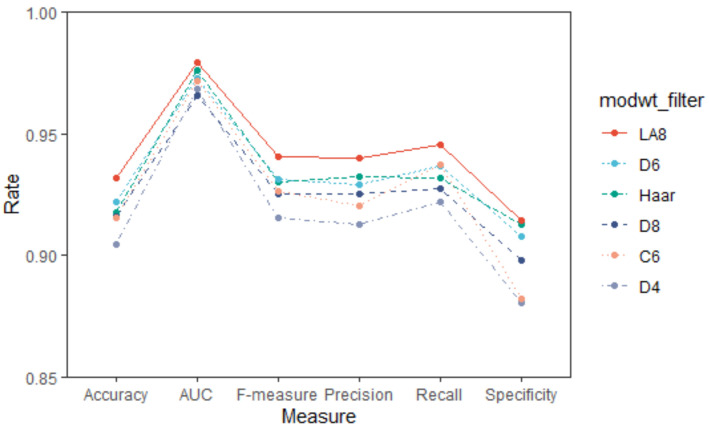
AD vs. NC discriminant results of the six wavelet filters based on LDA (mean value).

**Figure 3 healthcare-08-00476-f003:**
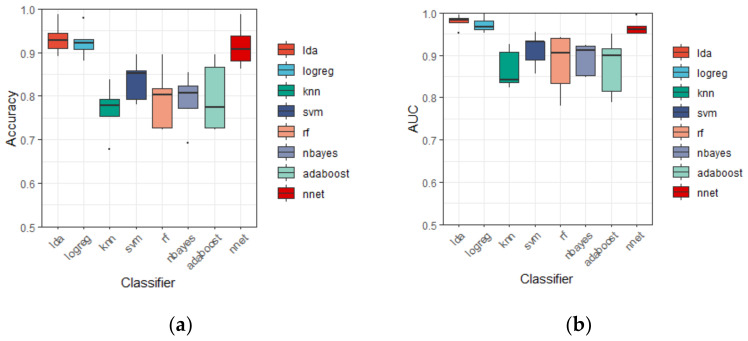
Boxplot of discriminant results by different classifiers using five-fold CV based on LA8 wavelet filter in terms of: (**a**) Accuracy; (**b**) AUC; (**c**) F-measure; (**d**) Specificity; (**e**) Recall; (**f**) Precision. Note: Each boxplot is constructed of the box and the whiskers. The box is drawn from *Q*_1_ (25th percentile) to *Q*_3_ (75th percentile) with a horizontal line drawn in the middle to denote the median. The upper whisker extends to the largest value no further than 1.5 × *IQR* from the *Q*_3_ (where *IQR* is the inter-quartile range, *IQR* = *Q*_3_ − *Q*_1_). The lower whisker extends to the smallest value at most −1.5 × *IQR* from the *Q*_1_. The dots plotted individually are outliners whose values are beyond the end of whiskers (±1.5 × *IQR*).

**Figure 4 healthcare-08-00476-f004:**
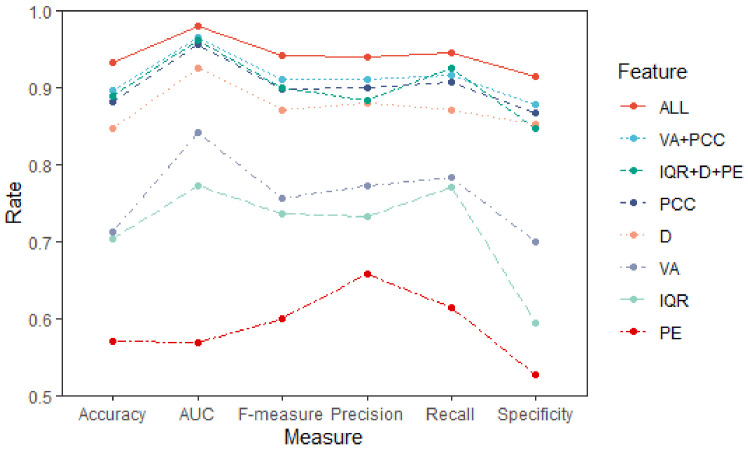
AD vs. NC discriminant results of the different features based on LDA (mean value).

**Figure 5 healthcare-08-00476-f005:**
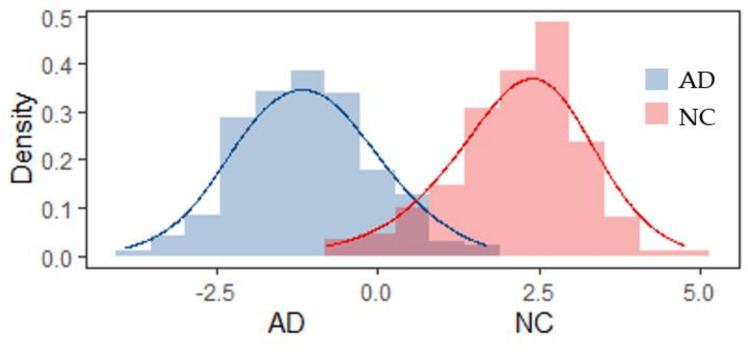
Distribution of linear combination of the features of AD or NC after LDA.

**Table 1 healthcare-08-00476-t001:** The demographic characteristics of participants.

Subjects	*N*	Sex/*N* (%)		Age/Years	
		Female	Male	Median (*Q*_L_, *Q*_U_)	Range
HC	23	14 (60.9)	9 (39.1)	59 (50, 74)	(44, 83)
AD	23	12 (52.2)	11 (47.8)	73 (56, 80)	(49, 85)

**Table 2 healthcare-08-00476-t002:** The maximum decompose layer for different wavelet filters.

Wavelet Filters	Haar	D4	D6	D8	LA8	C6
Maximum decompose layer	10	8	7	7	7	7
The number of layers used	6	4	3	3	3	3
The number of features extracted	1148	1104	832	832	832	832

**Table 3 healthcare-08-00476-t003:** Discriminant results by different classifiers using five-fold CV based on LA8 wavelet filter (mean ± standard deviation, %).

Classifier	Accuracy	AUC	F-Measure	Specificity	Recall	Precision
LDA	93.18 ± 3.65	97.92 ± 1.66	94.06 ± 4.04	91.45 ± 6.98	94.55 ± 3.85	94.02 ± 7.95
Logreg	92.44 ± 3.61	97.18 ± 1.77	93.36 ± 4.31	88.04 ± 9.71	94.26 ± 5.07	93.05 ± 8.32
KNN	76.76 ± 5.85	86.69 ± 4.61	78.79 ± 9.21	75.81 ± 2.76	76.86 ± 12.06	81.85 ± 10.45
SVM	83.56 ± 4.81	91.28 ± 3.99	84.92 ± 7.74	81.18 ± 7.78	85.53 ± 11.04	85.85 ± 11.43
RF	79.28 ± 7.18	88.05 ± 7.12	82.01 ± 9.49	66.30 ± 8.43	87.04 ± 11.07	78.78 ± 12.77
Nbayes	78.97 ± 6.22	89.12 ± 3.81	77.28 ± 13.11	85.68 ± 9.14	69.26 ± 16.99	88.88 ± 6.89
Adaboost	79.73 ± 8.03	87.40 ± 6.85	81.69 ± 10.21	75.41 ± 11.00	82.89 ± 11.40	82.00 ± 13.85
NNet	91.47 ± 4.93	96.60 ± 1.84	92.68 ± 5.18	85.02 ± 12.13	94.34 ± 5.49	91.76 ± 9.50

**Table 4 healthcare-08-00476-t004:** Discriminant results by different features using five-fold CV and LDA based on LA8 wavelet filter (mean ± standard deviation, %).

Feature	Accuracy	AUC	F-Measure	Specificity	Recall	Precision
*VA*	71.32 ± 7.25	84.13 ± 9.55	75.67 ± 8.57	70.04 ± 18.85	78.26 ± 12.57	77.32 ± 19.06
*PCC*	88.18 ± 5.28	95.65 ± 2.82	89.87 ± 3.99	86.67 ± 5.45	90.66 ± 7.79	89.89 ± 7.35
*IQR*	70.32 ± 9.38	77.20 ± 14.69	73.61 ± 13.37	59.51 ± 19.71	76.99 ± 17.65	73.26 ± 15.15
*D*	84.75 ± 3.77	92.58 ± 5.53	87.03 ± 2.81	85.18 ± 6.48	87.05 ± 7.38	88.02 ± 8.46
*PE*	57.09 ± 19.75	56.82 ± 21.36	59.99 ± 21.81	52.65 ± 23.68	61.36 ± 34.15	65.82 ± 18.38
*VA* + *PCC*	89.52 ± 6.32	96.43 ± 2.30	91.03 ± 5.74	87.80 ± 6.66	91.65 ± 6.44	90.98 ± 8.77
*IQR* + *HCC* + *PE*	88.83 ± 4.18	96.16 ± 2.29	89.97 ± 5.30	84.70 ± 4.31	92.45 ± 6.13	88.25 ± 9.06
ALL	93.18 ± 3.65	97.92 ± 1.66	94.06 ± 4.04	91.45 ± 6.98	94.55 ± 3.85	94.02 ± 7.95

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
