# Peer review of "A Robust Discriminant Framework Based on Functional Biomarkers of EEG and Its Potential for Diagnosis of Alzheimer’s Disease"

_healthcare, 2020, doi:10.3390/healthcare8040476_

Round 1
Reviewer 1 Report
I think the authors present some interesting data, and the framework they describe using the EEG signals clearly discriminates normal elderly controls from Alzheimer patients. However, the number of patients and groups used are very low. No patients with dementia were used, neither patients with early Alzheimer diagnosis. Moreover, I think it is not clear what the authors have done differently from other’s, since there are a lot of approaches similar to this one. Where is the novelty? Where is the comparation with other similar approaches? Finally, as the authors assume in the last part of the discussion, I think at least some of this drawbacks should be addressed in order to have a more complete and valuable work, before publication.
- Main findings of the study:
. The authors wanted to find robust EEG biomarkers and propose a systematic discrimination framework based on signal processing and computer-aided techniques to distinguish Alzheimer (AD) patients from normal elderly controls (NC);
.The authors processed the EEG signals by Wavelet Transform (WT), an effective time-frequency analysis. Different wavelet features were extracted and analyzed to reach automatic identification of AD and NC patients/controls.
. Authors claim that they have developed a discrimination framework that can identify AD from NC with high performance in a systematic routine.
- Limitations
. The number of participants should be higher, 23 is very low, although I understand it is a difficult process to enroll patients and controls to these studies. Authors should get other working cohorts.
. Another important limitation is the fact that no males were included, although the female preponderancy to Alzheimer
. Other important limitation is the absent of a group of patients without Alzheimer, but with dementia, since this is a pathology that is important to discriminate form Alzheimer. Authors say that patients with these pathologies were excluded…it would have been a richer work, if they were included, in my view.
. Author claims that “Early diagnosis is crucial for optimal treatment planning and may avoid rapid deterioration of the illness” so it would have been a plus to have two AD patient’s groups, with early diagnosis and with the disease more developed. I only find a Alzheimer patient group with the disease already advanced.
. The novelty of the work in not clear, compared to other recent published papers, e.g. https://journals.plos.org/plosone/article?id=10.1371/journal.pone.0193607
Major revisions:
. All the points raised in Limitations section that refer to important issues. They have to be addressed, and corrected, when possible.
. I could not find clearly a comparation between this pre-processing technique and others already developed. And this is important, since this is the core of the paper, and has been under research in the last decade (Curr Alzheimer Res. 2010 Sep;7(6):487-505)
. The description of the groups pf patients is insufficient. How they were selected? Based in which tests/inquiries? Alzheimer’s patients, level of the disease?
. The authors claim that this framework could be used in a systematic routine. So, a scheme of the channels and post treatment used in this study compared to other studies should be clarified
. The authors acknowledge some of the described limitations, since they describe them in the final conclusions. However, some of them should be addressed, for the paper to be relevant. E.g,… after training the model, early diagnosed AD patients as well as patients with dementia should have be tested, to understand if the model can also distinguish them from healthy controls and AD patients (in the case of dementia patients). Only after this approach the model is ready to be tested in other cohorts.
Minor points:
. A final cartoon/graphical abstract/scheme would be a plus to the paper
Reviewer 2 Report
Thank you for the opportunity to review the manuscript entitled "A Robust Discriminant Framework Based on functional biomarkers of EEG for diagnosis of Alzheimer’s Disease” in Healthcare. This work represents an important area of inquiry that is relevant to the readership of this journal.
The statistical analyses were well conducted and represents the principal strength of the work. I applaud the use of R software
This is an interesting manuscript, but I have several issues the authors should address before publication can be considered:
- Why authors did not report Youden Index?
- The analysis of modifiable risk factors of dementia has been put at stake in a recent work (Livingston et al., 2020. DOI: https://doi.org/10.1016/S0140-6736(20)30367-6). Please, comment.
Author Response
Response to Reviewer 2 Comments
Point 1: Why authors did not report Youden Index?
Response 1: Thank you very much for your advice. In the process of the classification framework developing, six evaluation indexes such as AUC were used to compare classification performance of different models, and the best model was selected based on these indexes. However, in the perspective of model application, it is important to select a suitable threshold based on maximum Youden index. Due to the five-fold cross validation used in this study, a consistent optimal cut-off value cannot be determined, so we did not report the Youden index, AUC was used to evaluate the overall discriminant effect of the model.
Point 2: The analysis of modifiable risk factors of dementia has been put at stake in a recent work (Livingston et al., 2020. DOI: https://doi.org/10.1016/S0140-6736(20)30367-6). Please, comment.
Response 2: Thank you very much for your recommendation. We have learned a lot from the article. Alzheimer's disease is a heavy burden on families and society. And Alzheimer's disease is irreversible, primary prevention is the fundamental and the only measure to prevent it for now. The analysis of modifiable risk factors of dementia is very important for preventing the dementia before it occurs, and it could save more families using less cost. Therefore, the study and discovery of risk factors is beneficial for the whole society in long-term. In addition, for those who already have dementia, early detection, early diagnosis and early treatment are key steps to slow the progression of the disease and improve the life quality of the patients. The study aims to explore diagnostic markers and contribute to the detection of Alzheimer's disease.
Round 2
Reviewer 1 Report
I think the authors answered, quite well, to all the points raised in the revision. However, I think they should have also incorporated some of that doubts and explanations into the manuscript, including what this work brings new, compared to others, already published.
Point 7 in the answer regarding the description of the groups pf patients, should be much better detailed. And not as it was changed in the manuscript ( line 113 "...and neuroimaging tools, etc.").
The title should have the expression "potential diagnosis", since no early diagnosed AD patients as well as patients with dementia were tested, to understand if the model can also distinguish them from healthy controls and late AD patients (that what was evaluated).
A graphical abstract is still missing, in my view.
Author Response
thank you very much for your suggestion!
Point 1: I think the authors answered, quite well, to all the points raised in the revision. However, I think they should have also incorporated some of that doubts and explanations into the manuscript, including what this work brings new, compared to others, already published.
Response 1: Thank you very much for your suggestion. We added more contents in the manuscript (line 340, line 345, and line 351) to give more explanations about the main advantage of the study and the doubts of the study.
The details are: (line 340) “And the EEG with 16 channels used in this study were widely available in low-income and middle-income countries, making the proposed classification framework more practical.”, (line 345) “By segmenting the continuous EEG signals we got 631 epochs for analysis finally, and the cross-validation ensured the stability of the classification results.”, and (line 351) Most of the patients came to the First Affiliated Hospital of SYSU (one of the best hospitals in South China) for severe clinical symptoms, which resulted in the absence of the MCI in this study. The gradual realization of medical information sharing in the future could help us track the information of patients with Alzheimer's disease at early stages or before they become ill, thus promoting the early diagnosis of the disease. However, the classification framework proposed in the study based on the existing study are still potential for aiding the diagnosis of AD with high performance and can be easily reproduced on future data.
Other advantages of the study compared to others can be found in the manuscript: (line 161) “The MODWT eliminates this down-sampling and creates wavelet coefficients of which the length at each layer will be the same as the original time series, thus it is more applicable in most contexts.”, and (line 333) “The time-frequency analysis method of MODWT accompanying with LA8 wavelet better captured the characteristic features in the EEG signals of AD and thus well adapted to the classification task.”, and (line 337) “This is an inspiring result because the features selected in the study and classifier of linear discriminant analysis are easy to calculate and interpret,…”.
Point 2: Point 7 in the answer regarding the description of the groups pf patients, should be much better detailed. And not as it was changed in the manuscript ( line 113 "...and neuroimaging tools, etc.").
Response 2: We improved the description of the groups pf patients in the manuscript (line 112) “The diagnosed of AD patients were based on the Aging and the Alzheimer’s Association (NIA-AA) criteria []. The details were that: (1) the core clinical presentation of AD such as evident cognitive or behavioral (neuropsychiatric) symptoms were meet through history-taking and the Mini Mental State Examination (MMSE), and (2) neuroimaging tools such as the structural or functional (fMRI) scans were performed for detecting AD biomarkers and excluding other brain diseases, and (3) excluding patients with other forms of dementia or accompanying Parkinson's disease, epilepsy, psychiatric disorders and serious organic disease.”, and added the reference [31] which thoroughly described the diagnostic criteria used.
Point 3: The title should have the expression "potential diagnosis", since no early diagnosed AD patients as well as patients with dementia were tested, to understand if the model can also distinguish them from healthy controls and late AD patients (that what was evaluated).
Response 3: Thanks a lot for your advice. We have changed the title into “A Robust Discriminant Framework Based on functional biomarkers of EEG and its potential for diagnosis of Alzheimer’s Disease” (line 2 in the manuscript).
Point 4: A graphical abstract is still missing, in my view.
Response 4: Thank you for your suggestion. We are making the graphical abstract and need a little more time, we will submit it before November 9th.
This manuscript is a resubmission of an earlier submission. The following is a list of the peer review reports and author responses from that submission.